# Role of Menopausal Transition and Physical Activity in Loss of Lean and Muscle Mass: A Follow-Up Study in Middle-Aged Finnish Women

**DOI:** 10.3390/jcm9051588

**Published:** 2020-05-23

**Authors:** Hanna-Kaarina Juppi, Sarianna Sipilä, Neil J. Cronin, Sira Karvinen, Jari E. Karppinen, Tuija H. Tammelin, Pauliina Aukee, Vuokko Kovanen, Urho M. Kujala, Eija K. Laakkonen

**Affiliations:** 1Gerontology Research Center and Faculty of Sport and Health Sciences, University of Jyväskylä, 40014 Jyväskylä, Finland; sarianna.sipila@jyu.fi (S.S.); sira.m.karvinen@jyu.fi (S.K.); vukekovanen@gmail.com (V.K.); 2Neuromuscular Research Center, Faculty of Sport and Health Sciences, University of Jyväskylä, 40014 Jyväskylä, Finland; neil.j.cronin@jyu.fi; 3Department for Health, Bath University, Bath BA2 7AY, UK; 4Faculty of Sport and Health Sciences, University of Jyväskylä, 40014 Jyväskylä, Finland; jari.e.j.karppinen@jyu.fi (J.E.K.); urho.m.kujala@jyu.fi (U.M.K.); 5LIKES Research Centre for Physical Activity and Health, 40700 Jyväskylä, Finland; tuija.tammelin@likes.fi; 6Department of Obstetrics and Gynecology, Pelvic Floor Research and Therapy Unit, Central Finland Central Hospital, 40620 Jyväskylä, Finland; pauliina.aukee@ksshp.fi

**Keywords:** menopause, female aging, skeletal muscle, sarcopenia, estradiol, physical activity

## Abstract

In midlife, women experience hormonal changes due to menopausal transition. A decrease especially in estradiol has been hypothesized to cause loss of muscle mass. This study investigated the effect of menopausal transition on changes in lean and muscle mass, from the total body to the muscle fiber level, among 47–55-year-old women. Data were used from the Estrogenic Regulation of Muscle Apoptosis (ERMA) study, where 234 women were followed from perimenopause to early postmenopause. Hormone levels (estradiol and follicle stimulating hormone), total and regional body composition (dual-energy X-ray absorptiometry (DXA) and computed tomography (CT) scans), physical activity level (self-reported and accelerometer-measured) and muscle fiber properties (muscle biopsy) were assessed at baseline and at early postmenopause. Significant decreases were seen in lean body mass (LBM), lean body mass index (LBMI), appendicular lean mass (ALM), appendicular lean mass index (ALMI), leg lean mass and thigh muscle cross-sectional area (CSA). Menopausal status was a significant predictor for all tested muscle mass variables, while physical activity was an additional significant contributor for LBM, ALM, ALMI, leg lean mass and relative muscle CSA. Menopausal transition was associated with loss of muscle mass at multiple anatomical levels, while physical activity was beneficial for the maintenance of skeletal muscle mass.

## 1. Introduction

Skeletal muscle is responsible for movements under voluntary control, but it also has an important role in metabolism [1,2]. During aging, muscle mass decreases due to an imbalance in muscle protein turnover and cell atrophy [3]. In women, aging-related hormonal changes accelerate especially during menopause, which women face in middle age [4,5]. Estradiol, the main female sex steroid hormone lost due to menopause, has been proposed to be among the molecular regulators of female skeletal muscle properties [6]. Menopausal transition starts when the hypothalamus begins to show signs of aging and the release of gonadotropin-releasing hormone becomes unsynchronized [7]. This leads to an imbalance between the release of luteinizing hormone (LH) and follicle stimulating hormone (FSH), and when combined with simultaneous ovarian aging, estradiol levels decrease [7]. When menopause occurs, i.e., when the menstrual cycle ceases completely, systemic FSH levels are set high and estradiol levels low. These hormonal changes, particularly the loss of systemic estradiol from the ovaries, have been suggested to have an impact on whole body and skeletal muscle composition, whereby more adipose tissue is accumulated and total muscle mass decreases [8,9,10]. To date, the majority of human studies regarding menopause and muscle mass have been based on cross-sectional studies (e.g., [11,12]), and to our knowledge, the number of longitudinal studies [8,13,14] is limited.

Total muscle mass is dependent on the number and size of muscle cells. Human skeletal muscle consists of three major muscle fiber types, which differ in functional and metabolic properties. This distinction is based on the content of the dominant myosin heavy chain (MHC) type, the main determinant of muscle cell contractile properties. Type I myofibers (MHC-I) or “slow fibers” are myofibers with high mitochondrial and capillary density, and they provide power for long-term contractions, e.g., during postural maintenance and endurance sports. Type IIA (MHC-IIA) and type IIX (MHC-IIX) fibers or “fast fibers” are more glycolytic in nature, and contribute to short-duration high-intensity activities [15]. While the proportion of different fiber types in muscles is mostly attributable to genetics, it seems that age, level and type of physical activity, inactivity and body weight also have an effect [16,17]. Only a few studies have focused specifically on the role of estradiol in fiber type distribution [18,19,20], so the role of the main female sex hormone in this process remains largely unknown. 

The aim of this longitudinal study was to investigate how menopause affects lean and skeletal muscle mass in women. We were also interested in the possible association between physical activity level and skeletal muscle tissue properties during the menopausal transition. To gain a comprehensive insight into the menopausal transition, we used whole body, limb and cellular variables to estimate lean body mass (LBM), appendicular lean mass (ALM), thigh muscle cross-sectional area and *m. vastus lateralis* muscle fiber composition before and after menopause, as presented in Figure 1.

## 2. Materials and Methods

### 2.1. Study Design and Participants

This study utilized data and samples from the Estrogenic Regulation of Muscle Apoptosis (ERMA) study [21]. The flow chart of the study is presented in Figure 2. Postal invitations to participate were sent to 6878 randomly selected 47–55-year-old women from the Finnish National Registry who were living in the city of Jyväskylä or neighboring regions. Of these 6878, 3064 returned the pre-questionnaire. After applying the exclusion criteria of self-reported body mass index (BMI) over 35 kg/m^2^, use of estrogen-containing contraceptives or other medications and conditions affecting ovarian function, altered systemic inflammatory status and health concerns or conditions jeopardizing ordinary physical function, 1627 participants were invited to the laboratory for further assessment. Of these, 1393 came to the first blood sampling session and were assigned to menopausal groups (premenopausal, early perimenopausal, late perimenopausal and postmenopausal), based on their FSH levels and menstrual bleeding reported in a diary. The ERMA follow-up study (core-ERMA) invited 381 women assigned to the early or late perimenopausal groups to participate in a follow-up study over the menopausal transition (Figure 2). These core-ERMA participants were followed from perimenopause to early postmenopause (mean follow-up time 15.3 ± 8.6 months), with regular laboratory visits every 3–6 months. Follow-up visits were continued until FSH level was above 30 IU/mL and no menstrual bleeding was reported during the last five to six months. Within a few days of the detection of high FSH concomitant with lack of menstruation, the participant was asked to come to the laboratory for a control visit to re-test the FSH levels and re-check the bleeding diary. If in the control visit FSH levels were still high and there was still no menstrual bleeding, the participant was considered to be early postmenopausal and she was asked to come to the laboratory for final follow-up measurements. During the follow-up period, 234 women became postmenopausal and the final follow-up measurements repeating the baseline measurement protocol were performed. The number of participants who took part in the lean mass measurements used in the current study is presented in Figure 1. Eleven women did not participate in the dual-energy X-ray absorptiometry (DXA) measurements, therefore the final number of participants for total body composition measurements was 223. For the quantitative computed tomography (CT) scans and biopsies, all participants who at baseline did not use progestogen-containing contraceptives and had not had a hysterectomy were invited for measurements. The only exclusion criterion for DXA and CT was a previous history of cancer. Exclusion criteria for muscle biopsy were the use of blood thinning medications or hemophilia. During the follow-up, 37 women started using hormone replacement therapy (HT). The average follow-up time for these participants was 16.6 (6.4–35.6) months. At baseline, 21 participants in the HT-group were early perimenopausal and 16 were late perimenopausal. The most commonly used product was tablet form estradiol hemihydrate combined with dydrogesterone (Femoston, n = 15) and estradiol valeriate-containing tablets (Progynova, n = 7). Because HT use masks the progression of menopausal transition, HT users were invited to the final follow-up measurements approximately six months after their HT use was informed to the laboratory. This time period was considered to be sufficient to allow HT to have exerted its phenotypic effects. The duration of HT use varied from 2 to 337 days. During final follow-up measurements, one participant informed us that she had started using HT just two days prior to the laboratory visit. As she was already regarded to be postmenopausal, the final follow-up measurements were done as planned, but she was classified as an HT user. In the statistical analysis, the duration of HT use was controlled for all participants. All participants provided written informed consent prior to inclusion. The study was approved by the ethical committee of the Central Finland Health Care District (Dnro 8U/2014).

### 2.2. Hormone Measurements

Fasting serum samples were taken from an antecubital vein between 7–10 a.m. At baseline, women with a menstrual cycle were asked to come to the laboratory between cycle days 1 to 5. Serum estradiol (E_2_), follicle stimulating hormone (FSH), sulfated dehydroepiandrosterone (DHEAS) and sex hormone binding globulin (SHBG) levels were measured with IMMULITE 2000 XPi (Siemens Healthcare Diagnostics, UK).

After baseline measurements, participants started the follow-up. During follow-up, elevated FSH levels were checked with FSH control blood samples and after two similarly elevated levels combined with a lack of menstruation for at least 6 months, the participant was considered postmenopausal. The participant was then invited to the final follow-up visit for physiological measurements, during which the E_2_ and FSH levels were again measured. For the final analysis, the E_2_ and FSH values of the most recent follow-up visit and final follow-up visits were averaged to minimize the effect of daily fluctuations. Hormone levels for participants who started using HT were only obtained from the final follow-up visit.

### 2.3. Lean and Muscle Mass Measurements

Lean and muscle mass measurements were done after overnight fasting. Total LBM, appendicular lean mass (summed lean mass of arms and legs, ALM) and lean mass of the right leg were analyzed from DXA scans (LUNAR Prodigy; GE Healthcare, Chicago, IL). DXA measurement of right leg lean mass was chosen to accompany the quantitative computed tomography scans (CT) that were taken from the right thigh. LBM index (LBMI) was calculated by dividing LBM (kg) by height squared (m^2^). ALM index (ALMI) was calculated similarly with ALM and height squared. An ALMI cut off-limit of 5.67 kg/m^2^ [22] was used to detect sarcopenia. The right mid-thigh was scanned at the level of the muscle biopsy with CT (Siemens Somatom Emotion scanner, Siemens, Erlangen, Germany) (Figure 1). Total thigh cross-sectional area (CSA) and absolute and relative muscle areas were measured using appropriate thresholds in Python Software (version 3.6). From the cross-sectional image, the muscle portion including the femur was first separated using a machine learning algorithm called U-net [23] or manually if needed. Muscle cross-sectional area was separated from adipose tissue and bone by using Hounsfield unit (HU) limits for muscle. Relative muscle area was calculated by dividing absolute muscle area by total CSA. All images were analyzed using ImageJ Software (v.1.52, NIH) and Python.

### 2.4. Muscle Biopsies

Muscle biopsies were collected from the subpopulation of women at baseline and final follow-up who did not use progestogen at baseline and did not start using HT during the follow-up, i.e., those who went through the menopausal transition naturally. Biopsies were taken from the middle portion of the *m. vastus lateralis* by percutaneous needle biopsy under local anesthesia. All visible connective and adipose tissue was removed, and the sample was quickly divided into three parts. Two parts were assigned to biochemical and molecular biology analyses and were snap frozen in liquid nitrogen. The third part was embedded transversely on a cork with TissueTek and frozen in isopentane cooled in liquid nitrogen. All samples were stored at −150 °C until analysis.

### 2.5. Myosin Heavy Chain Isoform Separation with SDS-PAGE

Muscle samples assigned for protein analysis (weight ~ 4–12 mg) were homogenized in 1:100 myosin extraction buffer (0.1 M KCL, 0.1 M KH_2_PO_4_, 0.05 M K_2_HPO_4_·3H_2_O, 0.01 M EDTA, 0.02 M NaPPi, BME, Pepstatin A, Halt Proteinase and Phosphatase Inhibitor (ThermoFisher Scientific, Waltham, MA, USA)) with TissueLyser II (Qiagen, Germany). Homogenization was extended with 24-h shaking at +4 °C, followed by centrifugation for 10 min at 10,000× *g* at +4 °C (Eppendorf 5424, FA-45-24-11, Hamburg, Germany). Then, 20 µL of obtained supernatant was mixed with working Laemmli sample buffer and glycerol. Samples were heated for 4 min at +100 °C and then frozen to −20 °C. 200–300 ng of total protein was loaded into the SDS-PAGE gel system, consisting of 3% stacking gel and 6.7% separating gel with 30% glycerol. Electrophoresis was run in Bio-Rad Protean II Xi Cell for 42–44 h at 70–90 V at +4 °C. After the run, gels were fixed for one hour (40% ethanol, 10% acetic acid and 50% H_2_O) and washed with water. A sensitizer (0.02% sodiumthiosulphate in water) was applied to gels for 1 min. Gels were washed and incubated in cold 0.1% silvernitrate solution with formaldehyde. After staining, the gels were again washed and developed with 3% sodiumcarbonate solution, until the staining was visible. Developing was terminated in 5% acetic acid solution. Gels were imaged with ChemiDoc MP (v.2.2.0.08, Bio-Rad Laboratories, Inc., Hercules, CA, USA) and images analyzed with Image Lab (v.6.0.1, Bio-Rad Laboratories, Inc.). 

### 2.6. Myofiber Type Distribution and Size Measurement

Serial transverse sections of 10 µm in thickness were cut on a cryostat at −20 °C and attached to glass slides. Slides were air dried and fixed in 4% paraformaldehyde in PBS (pH 7.4). Samples were blocked with 5% goat serum and primary antibodies were added in 1% goat serum. Myosin heavy chain distribution was analyzed with mouse antibodies against type I or type II fibers (A4.74 (1:35) for type II and A4.951 (1:40) for type I, Developmental Studies Hybridoma Bank, University of Iowa, IA, USA). Rabbit antibody for laminin (L9393, 1:250, Sigma-Aldrich, St. Louis, MO, USA) was used to mark cell borders. Primary antibodies were incubated at +4 °C overnight and the next morning, the attachment of primary antibodies was visualized with fluorescent secondary antibodies (Alexa Fluor 546 for goat anti-mouse (A11003, 1:500) and Alexa Fluor 488 for goat anti-rabbit (A11008, 1:500), Invitrogen, Carlsbad, CA, USA). Sections were mounted with Prolong Gold with Dapi (P36931, Invitrogen) and imaged with confocal microscopy (LSM 700, Axio Observer, Zeiss, Oberkochen, Germany). Images were analyzed manually with ImageJ software (v.1.52, NIH). 

### 2.7. Physical Activity 

The intensity and volume of physical activity were evaluated using a structured physical activity questionnaire [24] and ActiGraph hip-worn accelerometers (GT3X+ or wGT3X+, Pensacola, FL, USA), as reported earlier [25]. The physical activity questionnaire was used to calculate metabolic equivalent (MET) index as the product of intensity*duration*frequency of activity to give a score of MET-hours per day [24]. The data analysis process for accelerometer measurements has been reported previously [26]. Briefly, the amount of time spent at different physical activity intensities was evaluated using triaxial vector magnitude cutoff points for light, moderate and vigorous physical activity intensities: 450, 2690 and 6166 counts per minute, respectively [25,27]. Moderate-to-vigorous physical activity (MVPA) was defined by computing the sum of moderate and vigorous physical activity. The participants wore the accelerometers for seven consecutive days during waking hours, leading to some variation in accelerometer wearing times between participants. Therefore, MVPA was normalized to 16-hour wearing time per day [28]. Longitudinal accelerometer data were obtained from 173 participants, because not all participants were willing to wear accelerometers, some devices were lost in return transit and some data were lost due to technical errors. 

### 2.8. Dietary Analysis

Diet quality score (DQS) was calculated based on a food-frequency questionnaire, which the participants completed at baseline and final follow-up measurements. The food-frequency questionnaire included 45 typical Finnish food items and 6 answer options. The DQS consisted of 11 elements characteristic of a healthy diet according to the Nordic Nutrition Recommendations 2012 (http://dx.doi.org/10.6027/Nord2014-002). The regular consumption of vegetables, fruits, and berries, dark or crispbread, low-fat dairy and fish, as well as nuts and seeds, was classified as beneficial. In contrast, a healthy diet was considered to only rarely include refined baked products, processed meats, sugary beverages, fast food, and sweet or salty snacks. Each component was worth 1 point, and the maximum score available was 11 points. A higher DQS score was considered to reflect a healthier diet. The DQS was partly adapted from [29]. Reconstruction of the original DQS was necessary, as our food-frequency questionnaire included some different food items and different wording of answers.

### 2.9. Background Variables 

Anthropometrics were measured after overnight fasting. Body mass was measured with a digital scale and height with a stadiometer. Body mass index (BMI) was calculated as body mass divided by height squared (kg/m^2^). Level of education was determined with a questionnaire and categorized as primary, secondary and tertiary. Data about smoking and alcohol consumption were collected with a structured questionnaire.

### 2.10. Statistical Analysis

Descriptive characteristics are reported as means and standard deviations (SD). All variables were evaluated for normality and parametric tests were used whenever possible. Independent samples *t*-test, chi-squared test and Mann–Whitney *U*-tests were used to compare baseline characteristics between early and late perimenopausal groups. Generalized estimating equations (GEE) tests of model effects was also used to study the possible differences between the menopausal groups in the longitudinal set-up. For longitudinal analyses, differences in lean and muscle mass variables between early and late perimenopausal groups, between baseline progestogen users and non-users, and between participants who did and did not start to use HT, were tested. Because no differences were found between the groups, they were combined for analysis. Paired *t*-test and Wilcoxon rank test were used to test for differences in lean and muscle mass variables between baseline and final follow-up. Pearson and Spearman correlations were calculated to examine associations between changes in physical activity level and lean mass measures, and variables from muscle biopsies. GEE-modelling was performed to examine more detailed associations between the change in lean and muscle mass measurements and covariates during the follow-up. To investigate associations between menopausal status and each of the lean and muscle mass variables, models were controlled for baseline progestogen use, differences in the duration of HT use, and follow-up time. Education was also tested as a possible predictor, but it failed to reach significance for all variables, so it was not included as a covariate. To further investigate whether physical activity and age were also significant predictors of some or all of the muscle mass variables, they were included in the model step by step. Statistical data analysis was carried out using IBM SPSS Statistics Software version 24 (Chicago, IL, USA), and a *P*-value ≤ 0.05 was considered statistically significant.

## 3. Results

### 3.1. Characteristics of the Population at Baseline

Age, demographics and systemic hormone levels are shown in Table 1. As expected, early perimenopausal women were younger than late perimenopausal women (*P* = 0.013). The early perimenopausal group had higher serum E_2_ levels than the late perimenopausal group (*P* < 0.001). Serum FSH levels were higher in the late perimenopausal group (*P* < 0.001). DHEAS and SHBG did not differ between the groups. Education level, smoking habits and alcohol consumption were similar in both menopausal groups. At baseline, more than half of the participants had natural bleeding status, meaning that they did not have conditions that could confound the detection of menstrual cycle, such as the use of intrauterine or other hormonal contraception, or hysterectomy. Notably, 94% of subjects in the early and 89% in the late perimenopausal group fulfilled the national recommendations of moderate-to-vigorous physical activity (at least 150 min of MVPA per week, ≈21 min per day) and can therefore be considered to be active. No differences were observed between the groups in accelerometer-measured or self-reported physical activity.

Table 2 shows that the early and late perimenopausal groups did not differ in anthropometry or any of the lean and muscle mass variables at baseline. Approximately 50% of the participants were normal weight according to BMI, and LBM was on average more than 50% of the total body mass. LBMI was similar in both groups. ALMI was similar in both groups, and of the whole perimenopausal group, 4.5% (four participants in the early and six in the late perimenopausal group) were sarcopenic at baseline based on the previously reported cut-off values [22]. Right leg lean mass was similar in the early and late perimenopausal groups. No differences were observed in relative or absolute muscle area between the groups at baseline.

### 3.2. Changes in Characteristics and Lean and Muscle Mass Variables during the Follow-Up

Because there were no differences in lean and muscle mass variables between the early and late perimenopausal groups at baseline, the groups were combined for the longitudinal analysis. Table 3 presents the characteristics and lean and muscle mass results from the whole group at baseline and final follow-up measurements. The duration from baseline to final follow-up was on average 465 (133–1323) days. Both total body mass and BMI significantly increased during the transition (*P* < 0.001 for both). Due to the pulsatile nature of estradiol and FSH during menopausal transition, the differences in these variables were analyzed with non-parametric tests. A significant change was observed for both hormones during the follow-up (*P* < 0.001), even when the HT-users were included. Significant decreases were seen in LBM (*P* = 0.019), LBMI (*P* = 0.018), ALM (*P* < 0.001), ALMI (*P* < 0.001), right leg lean mass (*P* = 0.002) and absolute (*P* < 0.001) and relative muscle CSA (*P* < 0.001), during the menopausal transition. No change was observed in DQS between baseline and final follow-up.

Progesterone has been suggested to affect female muscle function [30,31]. Therefore, to increase the robustness of our analysis, we separately investigated changes in lean and muscle mass measures among women who used progestogen-containing contraceptives at baseline (Appendix A) and women who did not (Appendix A). A similar analysis was also performed for non-HT users (Appendix A) and those who started using HT during follow-up (Appendix A). At baseline, progestogen users and non-users did not differ from each other regarding estradiol and FSH levels, physical activity or lean mass measurements. Only age statistically differed at baseline (*P* = 0.004), whereby non-progestogen users were 0.8 years older than users. Of the 73 participants who used progestogen-based medication at baseline, 44 reported still using only progestogen containing medication (IUC or tablets) at final follow-up, and 15 reported using estradiol + progestogen containing hormone replacement therapy at final follow-up. Thirteen participants reported not using any hormone contraception or hormone replacement therapy at final follow-up, one of whom had undergone hysterectomy. One participant did not answer this question at final follow-up, but she was regarded as a current progestogen-user at final follow-up measurements. During follow-up, no significant changes were seen in any of the lean or muscle mass variables in the baseline-progestogen users. A decrease was observed in estradiol (−37%, *P* = 0.002) and an increase in FSH (+82%, *P* < 0.001). Non-progestogen users at baseline had very similar results to those presented in Table 3 for the whole follow-up group. 

In the HT using subgroup, due to HT-use, the serum level of estradiol was higher at final follow-up when compared to baseline, but just failed to reach significance. FSH levels were also higher at final follow-up (*P* = 0.047). A significant decrease was only observed in absolute muscle area (*P* = 0.021), while LBM, LBMI, ALM, ALMI, right leg lean mass and relative muscle area remained unchanged during the transition. The lean and muscle mass results obtained from the sub-group of non-HT users did not differ from the results presented in Table 3, except for self-reported physical activity, which was significantly higher at final follow-up in the non-HT users (4.3 ± 4.0 vs. 4.6 ± 3.7 MET-hours/day, *P* = 0.043).

### 3.3. Longitudinal Associations Between Menopausal Status, Covariates and Lean and Muscle Mass Variables

A correlation analysis was performed between the change in physical activity measures and important lean mass measures. Variables were formed by subtracting the baseline value from the final follow-up value (Δvariable). The change in the level of accelerometer-measured physical activity (ΔMVPA) was weakly, yet significantly positively, associated with ΔLBM (r = 0.182, *P* = 0.027), ΔLBMI (r = 0.182, *P* = 0.027), ΔALM (r = 0.235, *P* = 0.004), ΔALMI (r = 0.238, *P* = 0.004) and Δright leg lean mass (r = 0.241, *P* = 0.003).

Longitudinal associations between menopausal status and different lean and muscle mass variables during the menopausal transition were also tested with GEE-model (Table 4). As GEE tests of model analysis did not reveal any categorical differences between groups discordant for HT or progestogen use, the whole perimenopausal group was combined and possible hormone use was added to the model. Calculated self-reported MET-hours/day values were used as a measure of physical activity, because data were available from a higher number of participants than for MVPA. However, the same analyses were also performed using accelerometer-measured MVPA and the results, which were very similar to those for self-reported MET-hours/day, are presented in Appendix A. For LBM and ALMI menopausal status, baseline use of progestogen and physical activity level measured in MET-hours/day were significant predictors (for all *P* ≤ 0.050). For LBMI, menopausal status, baseline use of progestogen and age were significant predictors (*P* ≤ 0.037 for all). For ALM, right leg lean mass and relative muscle area menopausal status and physical activity remained significant in the adjusted model (*P* ≤ 0.011 for all). For absolute muscle area, only menopausal status remained significant (*P* < 0.001) when the same explanatory models were used.

### 3.4. Changes at The Cellular Level

A subpopulation of participants gave biopsies at baseline and final follow-up (n = 25). They all underwent menopause naturally, as they did not use progestogen-containing contraception, nor did they start HT use during follow-up. Mean time between biopsy samples was 385 days (115–999 days). More specific information about the characteristics of these women is provided in Appendix A. Changes in lean and muscle mass variables during the follow-up did not differ between participants in the biopsied and non-biopsied groups (Appendix A).

Muscle biopsy samples assigned for protein analysis were used for myosin protein extraction. Relative myosin heavy chain proportion was analyzed with SDS-PAGE and silver-staining (Figure 3). Table 5 presents the proportion of different myosin isoforms, as percentages at baseline and final follow-up. 

No statistically significant differences were seen in the myosin isoform proportions when advancing through menopause. At baseline, type I myosin proportion was positively associated with the level of MVPA (r = 0.424, *P* = 0.035). At final follow-up, the proportion of type IIA myosin was negatively associated with ALM (r = −0.465, *P* = 0.019) and absolute muscle area (r = −0.521, *P* = 0.013). The proportion of type IIX myosin was positively associated with ALMI (r = 0.434, *P* = 0.030) and LBMI (r = 0.409, *P* = 0.042) at final follow-up.

Biopsies collected for immunohistological staining were used to analyze mean cross-sectional area and proportions of type I and II fibers. Figure 4 and Table 6 show the results of these analyses. 

Counted cell number per participant varied from 940 to 6300 cells. The size of individual muscle fibers was unchanged between baseline and final follow-up. On average, type II fibers were smaller than type I fibers at baseline and final follow-up (for both *P* < 0.001). No changes were seen in the fiber type ratios from baseline to final follow-up.

At baseline, a positive correlation was observed between type II fiber size and age (r = 0.786, *P* = 0.036). A higher proportion of slow fibers was also associated with higher single leg lean mass (r = 0.857, *P* = 0.014), appendicular lean mass (r = 0.964, *P* < 0.001) and body mass (r = 0.893, *P* = 0.007). At final follow-up, larger type II fiber size correlated with higher ALMI (r = 0.786, *P* = 0.036). Positive correlations were also found at postmenopause between type I fiber proportion and total lean mass (r = 0.893, *P* = 0.007) and body mass (r = 0.929, *P* = 0.003). No significant correlations were found between the amount of physical activity and muscle fiber size or fiber type proportion.

## 4. Discussion

This study showed significant declines in several lean and muscle mass parameters in a longitudinal study design spanning the menopausal transition. We were able to show decreases in total lean mass, appendicular lean mass and thigh muscle cross-sectional area at a range of anatomical levels that, to our knowledge, has not been studied and showed before. This study examined the amount of skeletal muscle in relation to menopausal status, as well as possible associations between physical activity level and skeletal muscle tissue during menopausal transition. The results highlight the importance of menopause-related hormonal changes in the loss of lean and muscle mass that appears to be independent of the effects of aging. Our results also highlight the moderate yet significant importance of physical activity in maintaining lean mass during middle age.

### 4.1. Menopausal Transition Decreases Lean and Muscle Mass

Studies of the association between menopausal status and muscle mass have mostly used cross-sectional study designs and compared different menopausal groups [11,12,32,33]. Together with a limited number of longitudinal studies [8,13,14], they have found menopause to be associated with a decrease in skeletal muscle mass. Our results are in line with these previous studies, and here we also show changes in skeletal muscle at more anatomical levels and by using a shorter timespan over the menopausal transition than has been used before. We found that the decrease in lean mass occurs between peri- and early postmenopause, which was reflected in total body and appendicular measures. The results obtained by DXA and CT imaging collectively indicated a 0.5%–1.5% reduction in muscle mass due to menopausal transition, regardless of whether muscle mass was assessed at the anatomical level of the whole body, the limbs or the thigh.

In this study, participating women were not allowed to use any estradiol-containing medication at baseline, but we did not exclude those who began using HT during the follow-up. HT use was controlled in the main analysis and the inclusion of HT users did not affect the results of this study. However, when examining HT users and non-users separately, we did not find a similar, significant decrease in lean mass measures during the transition as with the non-HT-using subpopulation. This is in line with previous literature presenting data about the protective role of HT in muscle loss [34,35]. Our study was not designed to investigate the effects of HT, so our results in this respect need to be interpreted with caution. In our study, the participants were heterogenous in their HT supplementation methods, dosages and usage times, as we did not control their hormone use. Furthermore, the number of HT-participants was relatively low (n = 37), which may have left our study underpowered to detect the effects of HT use. Since recent literature, as reviewed by Javed et al. [36], also presents conflicting results about the protective role of HT on muscle mass even after long-term use, clearly more studies are needed to resolve this issue.

The systemic levels of another abundant female sex hormone- progesterone- also decrease during menopause, but the role of progesterone in lean and muscle mass maintenance is less studied. Progesterone has been shown to increase protein synthesis in postmenopausal women [31], and beneficial neuronal properties of progesterone are well established, especially when combined with estradiol [30,37]. Here, we included women who used progestogen contraception at baseline and the use was controlled as “yes” or “no” in the main analysis. CT scans and muscle biopsies were taken from women who did not use progestogen at baseline. In our analysis, in the GEE-model progestogen use remained significant for multiple lean mass variables, and we did not observe a similar decrease in lean mass variables during the menopausal transition for the progestogen-using subgroup as for the non-users. Unfortunately, we did not have information about the exact duration of progestogen use for all participants, so assessing the specific role of progesterone in lean mass maintenance is difficult. That being said, progesterone might have beneficial effects on muscle mass maintenance during menopause, possibly due to increased muscle protein synthesis.

### 4.2. Associations of Menopausal Transition at the Single Cell Level

Aging has been shown to decrease muscle fiber area and there is evidence, albeit conflicting, of a shift in the muscle fiber type ratio towards a slower or more hybrid phenotype [38,39,40]. Most studies have reported that aging causes more visible changes in the size of type II fibers, while the size of slow type I fibers stays relatively unchanged in both sexes [41,42,43,44]. The role of aging in muscle tissue myosin expression has been previously studied in women in one cross-sectional study that compared 30- and 68-year old women, and the results also suggested a shift toward a slower phenotype in the older group [40]. Studies of the associations between menopause and skeletal muscle fiber size, type and myosin isoform expression are scarce. A few studies have examined the role of estrogen in animals, but the results are conflicting [19,45,46]. According to animal studies, a loss of ovarian function leads to a decline in myosin function in mature female mice [47], suggesting that myosin isoform availability may be functionally relevant. In humans, only a few studies have examined the response of skeletal muscle fibers to HT and revealed no difference between users and non-users [18,48]. To our knowledge, prior to the present study, no longitudinal studies have examined the myosin isoform distribution and muscle fiber size of perimenopausal and early postmenopausal women without HT use. 

During the menopausal transition, we did not observe a change in the cross-sectional area of either type I or type II fibers, but the two fiber types did differ from each other in size both at baseline and at final follow-up. The cross-sectional area of type I fibers was larger at both time points and remained rather unchanged, similarly to what has been reported cross-sectionally in a similar age group with differing hormonal statuses [18]. The aging-related reduction in the CSA of type II fibers that has been found in earlier studies [39,49,50] was not phenocopied in our study. This suggests that muscle fiber size is not influenced by menopause, although it is also possible that changes in cell size were not detected, due to the short follow-up duration or the limited number of participants. Participants in this study had about a 50/50 ratio of type I and II fibers, which is more evenly distributed than has been previously observed in women in the same age range [18]. Widrick et al. reported that 49–57-year-old postmenopausal women had a fiber ratio of 38/62, with no difference between HT users and non-users. In our longitudinal study, it may be that type II fibers were lost early in the menopausal transition and thus that reductions were no longer detectable during the transition from perimenopause to postmenopause. However, we consider it more likely that changes in fiber type ratio occur later during aging than immediately concomitant to menopause. A higher proportion of type II fibers in muscles might have more drastic effects on muscle mass, if type II fibers are more affected by aging. Studies of the relationship between fiber types and muscle mass have in fact reported a positive correlation between type II fiber proportion and skeletal muscle mass, in both men and women [39,40]. These results differ from our results, most likely because our study population was smaller and only included women.

### 4.3. Physical Activity Helps to Maintain Lean and Muscle Mass during Menopause

Here, we showed that while menopausal transition was a strong predictor of decreased lean and muscle mass, physical activity was positively associated with lean mass. Previous studies have shown that staying physically active during aging could potentially slow down the changes in skeletal muscle tissue caused by aging [51]. Physical activity has also been shown to be associated with female reproductive factors and menopausal symptoms [25], and together with estradiol, it seems to preserve favorable skeletal muscle properties [11,35,52]. We measured physical activity in two different ways; self-reports via a questionnaire, which measures mostly commute and leisure time activity, and 7-days of hip-worn accelerometer data, which included all wear-time activity, especially activities including steps. The results regarding the role of physical activity were similar with both measurements, which highlights the reliability of the results.

The combined effects of muscle fiber size/type and physical activity have not been widely studied in women, but it seems that older women are as capable of responding to both endurance and resistance exercise as males of the same age [43,53,54]. Subjects in our study remained physically relatively active and healthy during the transition, which might have positively affected their muscle fiber maintenance, and could therefore explain why no obvious decrease was seen in fiber size. No differences in physical activity levels were observed between the participants whose biopsies were used for fiber cross-sectional area analysis and non-biopsied subjects. In a recent study, the use of estradiol and progestogen-containing oral contraceptives combined with training was associated with larger type I CSA [20]. Moreover, after the training period, the fiber type shift from IIX to IIA was larger in women who used oral contraceptives, which, in contrast to our study, points to the role of estradiol in muscle cell modifications. We only analyzed muscle biopsies from women who went through natural menopause, without exposure to female hormones via oral contraceptives or HT, so they experienced fluctuating estradiol levels during the transition. We did not find associations between the level of physical activity and single muscle fiber size.

### 4.4. Strengths and Limitations

One of the limitations of this study is the relatively short follow-up time, especially after menopause. Although we carefully monitored the menopausal transition with sequential hormone measurements and menstrual bleeding diaries, we might have classified some women as still being perimenopausal at baseline or as postmenopausal too early, due to the relatively short follow-up time and typical fluctuations in hormone levels during the transition. Furthermore, although the study was intended to be observational, some of the participants might have increased their physical activity level during the follow-up, due to participation in this study and increased self-consciousness, which might have affected the outcomes when comparing only two timepoints. Whether or not this was the case, we were still able to see a clear decrease in lean mass at multiple body levels, which emphasizes the essential role of menopausal transition in the decline of lean and muscle mass. 

This study also has several strengths. One of the main strengths is the longitudinal design, which was conducted based on a personalized timetable. As well as being a limitation, the relatively short follow-up time may be considered as a considerable strength, as we were able to show significant changes already on this time scale, emphasizing the role of menopause over purely aging-related effects. The number of participants remained relatively high throughout the study, even though the follow-up period was not long enough for all of the original core-ERMA participants to reach early postmenopause before the end of the study. It may also be considered as a strength that we utilized several parameters from different anatomical levels to examine possible changes in muscle mass. We used DXA and CT imaging, which are highly accurate and repeatable and thus recommended methods for lean and muscle mass assessment at the whole body and limb levels. We also took muscle biopsies to examine potential cellular level changes. We had a reasonable number of repeated samples (n = 25) for myosin isoform expression analysis, but we were only able to do immunohistological staining for seven peri- and early postmenopausal women. Therefore, we cannot totally exclude the possibility that small sample size may have affected our results. When interpreting the results, it is also important to note that both muscle fiber CSA and fiber/myosin isoform proportions are known to be affected by the site of biopsy [41,44], and this might also have affected our results. Although we were obviously not able to obtain the follow-up biopsy from the exact same location as at baseline, an effort was made to take it from the closest proximity (~1 cm apart). Furthermore, multiple sections and all intact cells in them were counted to gain more data from the immunohistological samples. In addition, we took care that the CT scans were taken from the same location as the biopsies, to maximize the representativeness of these two completely different measurements.

## 5. Conclusions

This longitudinal study showed a significant decline in total body lean mass, appendicular lean mass and absolute and relative muscle cross-sectional area during the menopausal transition, suggesting an important role of female sex hormones in loss of muscle mass in women. Menopausal transition seems to have a role in loss of muscle mass that is independent of aging. Physical activity was associated with the maintenance of muscle mass during middle age, suggesting that women should stay physically active in order to reduce the risk of muscle mass loss-related symptoms, such as sarcopenia. Because muscle tissue is important not only for locomotion, but also for thermoregulation and whole-body metabolism, the menopause-related reductions in muscle mass demonstrated here may represent the onset of widespread negative effects on women’s health.

## Figures and Tables

**Figure 1 jcm-09-01588-f001:**
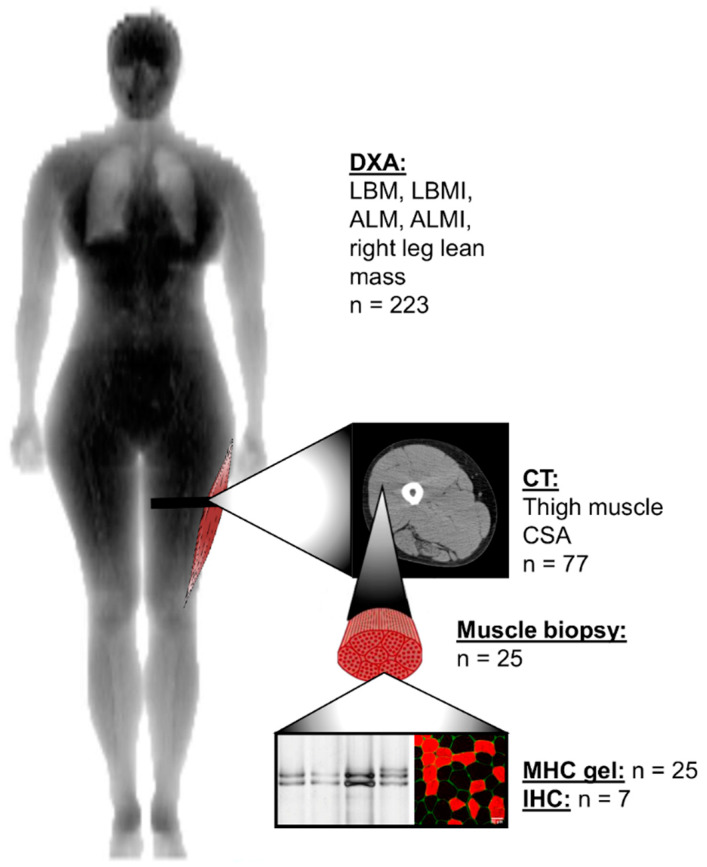
Anatomical levels of lean and muscle mass measurements and the number of the same participants measured at baseline and final follow-up time-points. ALM, appendicular lean mass; ALMI, appendicular lean mass index; CSA, cross-sectional area; CT, computed tomography; DXA, dual-energy X-ray absorptiometry; IHC, immunohistochemistry; LBM, lean body mass; LBMI, lean body mass index; MHC, myosin heavy chain.

**Figure 2 jcm-09-01588-f002:**
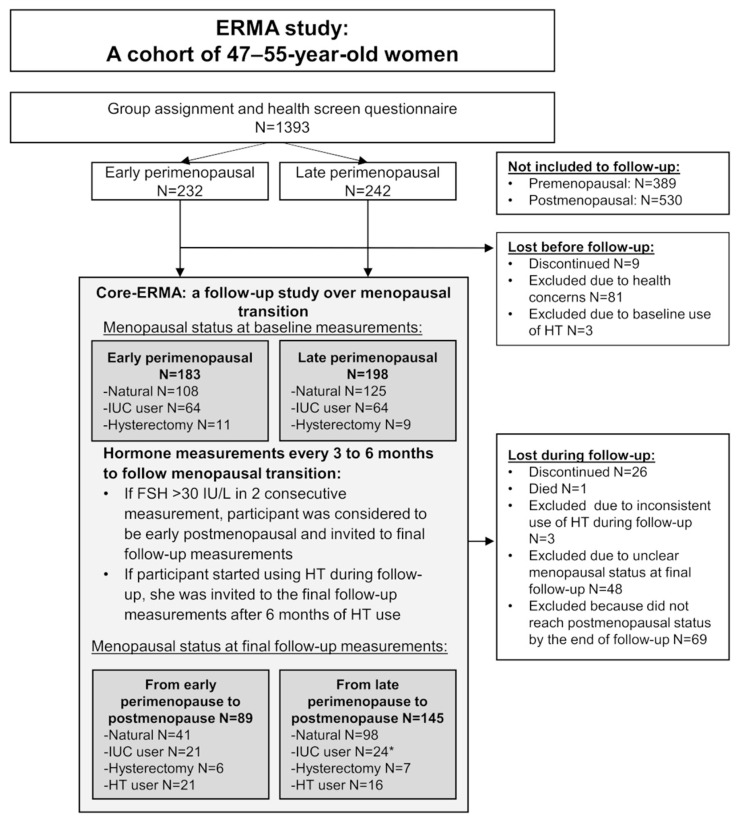
Flow chart of the longitudinal Estrogenic Regulation of Muscle Apoptosis (ERMA) study. FSH, follicle stimulating hormone; HT, hormone replacement therapy; IUC, intra-uterine hormonal contraceptive. * One former IUC user did not inform us if her IUC-status had changed during follow-up, so she was still counted as an IUC user at final follow-up measurements. IUC-group also includes women using progestogen pills.

**Figure 3 jcm-09-01588-f003:**
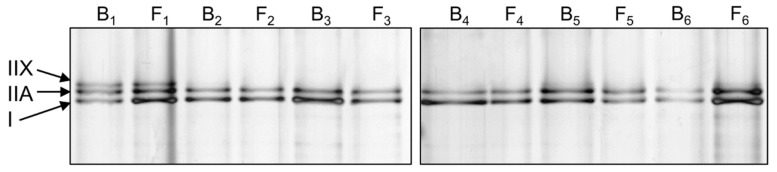
Representative silver-stained myosin heavy chain (MHC) SDS-PAGE results from six participants (1–6, B, baseline; F, final follow-up).

**Figure 4 jcm-09-01588-f004:**
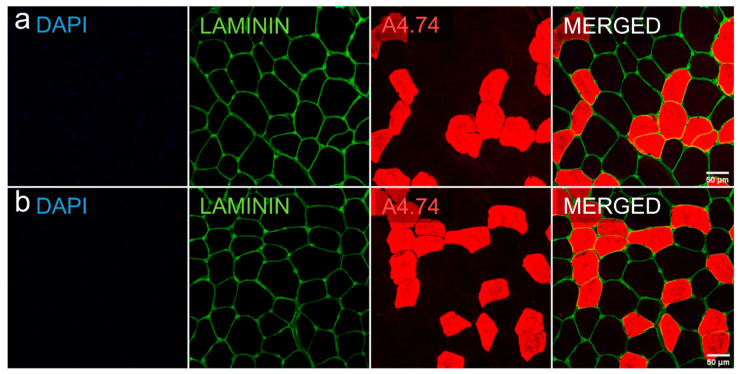
Representative cross-sectional area and fast myosin-based fiber typing of a muscle biopsy with Dapi (blue), laminin (green) and A4.74 (red, type II)-staining. Row (**a**) baseline, row (**b**) final follow-up.

**Table 1 jcm-09-01588-t001:** Characteristics of the perimenopausal ERMA baseline population (n = 234).

	Early Perimenopausal (n = 89) ^#^	Late Perimenopausal (n = 145) ^¤^	*P*
Age, y	51.2 ± 2.0	51.8 ± 1.8	**0.013 ^a^**
E_2_, nmol/L	0.46 ± 0.34	0.26 ± 0.18	**<0.001 ^b^**
FSH, IU/L	18.3 ± 5.0	46.9 ± 20.6	**<0.001 ^b^**
DHEAS, µmol/L	2.64 ± 1.42	2.91 ± 1.36	0.135 ^b^
SHBG, nmol/L	58.5 ± 24.3	53.1 ± 22.3	0.140 ^b^
Education level Primary Secondary Tertiary	1.1% 59.1% 39.8%	3.4% 47.6% 49.0%	0.261 ^b^
Smoking Never Quitter Current smoker	74.2% 23.6% 2.2%	66.9% 24.1% 9.0%	0.159 ^b^
Alcohol use, units/week	4.5 ± 4.5	3.9 ± 3.2	0.528 ^b^
Bleeding status Natural IUC Hysterectomy	56.2% 34.8% 9.0%	66.9% 29% 4.1%	0.151 ^c^
Physical activity			
MVPA, min/day ^X^	53.6 ± 22.8	52.0 ± 32.9	0.195 ^b^
MET-hours/day ^XX^	4.6 ± 4.2	4.3 ± 3.8	0.743 ^b^

Values are given as mean ± SD or as percentage. DHEAS, sulfated dehydroepiandrosterone; E_2_, estradiol; FSH, follicle stimulating hormone; IUC, intra-uterine contraception/progestogen use; MET, metabolic equivalent; MVPA, moderate-to-vigorous physical activity; SHBG, sex hormone binding globulin.^#^ In the early perimenopausal group, there are missing data regarding education, alcohol use and MET-h/day (n = 1) and regarding MVPA (n = 21). ^¤^ In the late perimenopausal group, there are missing data regarding DHEAS and SHBG (n = 1) and MVPA (n = 21). ^a^ Independent samples *t*-test, ^b^ Mann–Whitney *U*-test, ^c^ Chi-squared test, ^X^ accelerometer-measured, ^XX^ self-reported. Significant results (*P* ≤ 0.050) are shown in bold.

**Table 2 jcm-09-01588-t002:** Anthropometry and lean and muscle mass variables of perimenopausal women in the ERMA baseline population (n = 234).

	Early Perimenopausal (n = 89) ^#^	Late Perimenopausal (n = 145) ^¤^	*P*
Body mass, kg	69.2 ± 11.9	70.2 ± 10.8	0.534 ^a^
Body height, cm	165.2 ± 5.6	165.0 ± 5.8	0.771 ^a^
BMI, kg/m^2^ Underweight (<18.5) Normal weight (18.5–24.99) Overweight (25.0–29.99) Obese (>30)	25.4 ± 4.2 0% 57.3% 28.1% 14.6%	25.8 ± 3.8 0% 47.6% 36.6% 15.9%	0.287 ^b^
DXA-measurements			
LBM, kg	42.3 ± 4.8	41.4 ± 4.1	0.141 ^a^
LBMI, kg/m^2^	15.4 ± 1.4	15.2 ± 1.2	0.204 ^a^
ALM, kg	18.2 ± 2.4	17.9 ± 2.1	0.416 ^a^
ALMI, kg/m^2^	6.6 ± 0.7	6.6 ± 0.6	0.553 ^a^
Right leg lean mass, kg	6.8 ± 0.9	6.8 ± 0.8	0.494 ^a^
Computed tomography	(n = 24)	(n = 53)	
Absolute muscle area, cm^2^	166.1 ± 8.1	167.3 ± 10.3	0.636 ^a^
Relative muscle area, %	69.3 ± 4.2	69.8 ± 6.1 *	0.722 ^a^

Values are given as mean ± SD. ALM, appendicular lean mass; ALMI, appendicular lean mass index; BMI, body mass index; LBM, lean body mass; LBMI, lean body mass index. ^#^ In the early perimenopausal group there are missing data regarding DXA-measurements (n = 5). ^¤^ In the late perimenopausal group, there are missing data regarding DXA-measurements (n = 6). * Because of a technical failure in one CT scan, relative muscle area could not be calculated for n = 1 participant. ^a^ Independent samples *t*-test, ^b^ Mann–Whitney *U*-test.

**Table 3 jcm-09-01588-t003:** Characteristics and lean and muscle mass at baseline and final follow-up (n = 234).

	Baseline (Perimenopausal) n = 234	Final Follow-Up (Postmenopausal) n = 234	Difference %	*P*
Age, y	51.6 ± 1.9	53.0 ± 1.9	**+2.7**	**<0.001 ^a^**
Body mass, kg	69.8 ± 11.2	70.4 ± 11.6	**+0.9**	**<0.001 ^b^**
BMI, kg/m^2^	25.6 ± 4.0	25.8 ± 4.1	**+0.8**	**<0.001 ^b^**
E_2_, nmol/L	0.34 ± 0.27	0.24 ± 0.19	**−30**	**<0.001 ^b^**
FSH, IU/L	36.0 ± 21.6	66.9 ± 28.1	**+86**	**<0.001 ^b^**
DQS, points	5.7 ± 2.3	5.5 ± 2.2		0.207 ^a^
Physical activity				
MVPA, min/day ^X^ (n = 173)	51.8 ± 29.3	49.7 ± 23.6		0.567 ^b^
MET-hours/day ^XX^ (n = 231)	4.5 ± 3.9	4.7 ± 3.6		0.057 ^b^
DXA-measurements				
LBM, kg (n = 223)	41.7 ± 4.4	41.5 ± 4.4	**−** **0.5**	**0.019 ^a^**
LBMI, kg/m^2^ (n = 223)	15.3 ± 1.3	15.2 ± 1.3	**−** **0.7**	**0.018 ^a^**
ALM, kg (n = 223)	18.0 ± 2.2	17.8 ± 2.2	**−1.1**	**<0.001 ^a^**
ALMI, kg/m^2^ (n = 223)	6.6 ± 0.6	6.5 ± 0.6	**−1.1**	**<0.001 ^a^**
Right leg lean mass, kg (n = 223)	6.8 ± 0.9	6.7 ± 0.8	**−1.5**	**0.002 ^a^**
Computed tomography				
Absolute muscle area, cm^2^ (n = 77)	166.9 ± 9.6	165.3 ± 10.1	**−1.0**	**<0.001 ^a^**
Relative muscle area (%) (n = 76) *	69.6 ± 5.6	68.9 ± 6.0	**−1.0**	**<0.001 ^a^**

Values are given as mean ± SD. ALM, appendicular lean mass; ALMI, appendicular lean mass index; BMI, body mass index; DQS, diet quality score; E_2_, estradiol; FSH, follicle stimulating hormone; LBM, lean body mass; LBMI, lean body mass index; MET, metabolic equivalent; MVPA, moderate-to-vigorous physical activity. * n = 76: because of a technical failure in one CT scan, relative muscle area could not be calculated. ^a^ paired t-test, ^b^ Wilcoxon Signed rank test, ^X^ accelerometer-measured, ^XX^ self-reported. Significant results (*P* ≤ 0.050) are shown in bold.

**Table 4 jcm-09-01588-t004:** GEE-model with self-reported MET-hours per day as a measure of physical activity.

	Model 1		Adjusted Model	
	B	P	B	P
**LBM** Menopausal status Use of HT Use of progestogen Follow-up time MET-hours/day Age	−0.193 0.000 1.193 0.001 - -	**0.026** 0.833 0.057 0.503 - -	−0.203 0.000 1.263 0.001 0.053 0.109	**0.019** 0.836 **0.050** 0.420 **0.036** 0.506
**LBMI** Menopausal status Use of HT Use of progestogen Follow-up time MET-hours/day Age	−0.069 0.000 0.290 0.000 - -	**0.029** 0.696 0.110 0.426 - -	−0.073 0.000 0.386 0.000 0.017 0.122	**0.020** 0.753 **0.037** 0.183 0.054 **0.009**
**ALM** Menopausal status Use of HT Use of progestogen Follow-up time MET-hours/day Age	−0.231 0.001 0.555 0.001 - -	**<0.001** 0.362 0.061 0.304 - -	−0.238 0.001 0.557 0.001 0.038 0.025	**<0.001** 0.354 0.069 0.278 **0.009** 0.766
**ALMI** Menopausal status Use of HT Use of progestogen Follow-up time MET-hours/day Age	−0.085 0.000 0.143 0.000 - -	**<0.001** 0.386 0.083 0.200 - -	−0.088 0.000 0.175 0.000 0.014 0.046	**<0.001** 0.326 **0.039** 0.082 **0.005** 0.055
**Right leg lean mass** Menopausal status Use of HT Use of progestogen Follow-up time MET-hours/day Age	−0.088 0.000 0.214 0.000 - -	**0.001** 0.398 0.054 0.224 - -	−0.091 0.000 0.209 0.000 0.017 0.005	**0.001** 0.395 0.066 0.217 **0.011** 0.870
**Absolute muscle area *** Menopausal status Use of HT Follow-up time MET-hours/day Age	−1.597 −0.001 −0.001 - -	**<0.001** 0.811 0.833 - -	−1.586 −0.001 0.000 0.047 0.236	**0.001** 0.797 0.938 0.801 0.748
**Relative muscle area *** Menopausal status Use of HT Follow-up time MET-hours/day Age	−0.007 −1.5 × 10^−5^ −7.5 × 10^−6^ - -	**<0.001** 0.488 0.716 - -	−0.007 −6.7 × 10^−6^ −6.9 × 10^−6^ 0.002 −0.002	**<0.001** 0.721 0.747 **0.011** 0.667

Model 1: adjusted for menopausal status, HT use in days, baseline use of progestogen and follow-up time in days. Adjusted model: adjusted for menopausal status, HT use in days, baseline use of progestogen, follow-up time in days, MET-hours/day and age at baseline. * Absolute and relative muscle areas were not adjusted for baseline progestogen use, as all participants were non-users at baseline. ALM, appendicular lean mass; ALMI, appendicular lean mass index; HT, hormone replacement therapy; LBM, lean body mass; LBMI, lean body mass index; MET, metabolic equivalent. Significant results (*P* ≤ 0.050) are shown in bold.

**Table 5 jcm-09-01588-t005:** Relative proportions of different myosin isoforms in SDS-PAGE at perimenopause and early postmenopause.

Myosin Isoform	Baseline n = 25	Final Follow-Up n = 25	*P*
**Type I (%)**	50.6 ± 11.2	52.9 ± 8.5	0.619
**Type IIA (%)**	40.6 ± 9.6	41.4 ± 6.0	0.563
**Type IIX (%)**	8.8 ± 10.8	5.7 ± 9.8	0.116

Values are presented as mean ± SD. Wilcoxon Rank test was used for statistical analysis.

**Table 6 jcm-09-01588-t006:** Mean cross-sectional area and proportions of type I and II fibers measured with immunohistological stainings.

	Baseline n = 7	Final Follow-Up n = 7	*P*
Slow (type I) cells, µm^2^	3526 ± 1334	3525 ± 1618	0.735 ^a^
*Percentage*	53.3 ± 10.9	51.1 ± 14.4	0.398 ^a^
Fast (type II) cells, µm^2^	2098 ± 948	2399 ± 1218	0.128 ^a^
*Percentage*	46.7 ± 10.9	48.9 ± 14.4	0.398 ^a^
*P*-value for difference between cell types	**<0.001 ^b^**	**<0.001 ^b^**	

Values are presented as mean ± SD. ^a^ Wilcoxon rank test, ^b^ Mann–Whitney *U*-test. Significant results (*P* ≤ 0.050) are shown in bold.

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
