# Peer review of "Role of Menopausal Transition and Physical Activity in Loss of Lean and Muscle Mass: A Follow-Up Study in Middle-Aged Finnish Women"

_jcm, 2020, doi:10.3390/jcm9051588_

Round 1

Reviewer 1 Report

Since this study is a peri-menopausal study for middle-aged women, it is expected to provide an important basis for elucidating the importance of physical activity to prevent muscular atrophy caused by menopause through analyzing the relationship between muscle properties and physical activity. I think it is a well-written paper overall. Complementation of the following is required.

1. Why is my muscle fiber size not affected by menopause? Discussions on this will need to be added. Is there anything other than the reason for the small number of subjects and the short duration?

2. Analysis and discussion are considered to lack a more in-depth approach than valuable data related to muscle fibers. It is also possible to calculate the area per muscle fiber, but it would be better if detailed discussions on the results of muscle fiber were added.

3. It is thought that a detailed analysis of physical activity needs to be considered, for example, whether or not to participate in an exercise program or its details.

4. In the course of conducting the research, it is considered that the contents of dietary intake need to be considered, and it is considered that there is a need to add the contents of this part.

Reviewer 2 Report

The study provided useful information on the effects of menopausal transition on skeletal muscle in a large group of women. The study’s methodology and statistical analysis were in good quality, and the presentation of the study was clear and thorough. There are only some points to be addressed:

  1. The average follow-up time for these participants was 16.8 (6.5–36.1) months or 465 (133–1323) days. It seems like a wide range of follow-up time. How was the follow-up time determined?

  1. Fasting serum samples were taken between 7-10 am. How many hours of fasting were implemented? Any instructions on diet, exercise, etc. were given to participants prior to blood draw? For example, were participants allow to take HT prior to blood draw if they were on it?

  1. The authors compared many variables at perimenopausal and postmenopausal. What about the effects of the duration of menopausal transition on these variables? For example, although there were no differences at baseline for early and late perimenopausal, what about the changes in postmenopausal between the 2 groups?

  1. A subpopulation of participants gave biopsies at baseline and final follow-up (n = 25). Are there differences between the subpopulation group and the original group (n=234)? A repeated measure ANOVA would be good to compare the two groups at baseline and follow-up respectively.

  1. The study stated Baumgartner’s definition of sarcopenia, that is, ALMI cut off-limit of 5.67 kg/m2. How many women were classified as sarcopenia in this study? Please report it.

  1. There is an updated definition of sarcopenia which combines ALMI and grip strength/gait speed. What do you think of the relationships between muscular strength and power, hormone levels and muscular fiber types during the transition?
